# Study on the Luminescence Performance and Anti-Counterfeiting Application of Eu^2+^, Nd^3+^ Co-Doped SrAl_2_O_4_ Phosphor

**DOI:** 10.3390/nano14151265

**Published:** 2024-07-28

**Authors:** Zhanpeng Wang, Quanxiao Liu, Jigang Wang, Yuansheng Qi, Zhenjun Li, Junming Li, Zhanwei Zhang, Xinfeng Wang, Cuijuan Li, Rong Wang

**Affiliations:** 1Beijing Key Laboratory of Printing and Packaging Materials and Technology, Beijing Institute of Graphic Communication, Beijing 102600, China; wzpbigc@163.com (Z.W.); yuansheng-qi@bigc.edu.cn (Y.Q.); 2National Center for Nanoscience and Technology, CAS Key Laboratory of Nanophotonic Materials and Devices (Preparatory), Beijing 100190, China; 3The GBA Research Innovation Institute for Nanotechnology, Guangzhou 510700, China; 4Beijing Key Laboratory for Sensors, Beijing Information Science & Technology University, Beijing 100192, China; li@bistu.edu.cn; 5Yunnan Jiake Packaging Technology Co., Ltd., Yuxi 653100, China; zhang13930678816@163.com (Z.Z.); wangxinfeng88331@gmail.com (X.W.); licuijuan1982@163.com (C.L.); wangrong6355@126.com (R.W.)

**Keywords:** long afterglow, SrAl_2_O_4_:Eu^2+^, Nd^3+^, combustion method, screen printing, anti-counterfeiting

## Abstract

This manuscript describes the synthesis of green long afterglow nanophosphors SrAl_2_O_4_:Eu^2+^, Nd^3+^ using the combustion process. The study encompassed the photoluminescence behavior, elemental composition, chemical valence, morphology, and phase purity of SrAl_2_O_4_:Eu^2+^, Nd^3+^ nanoparticles. The results demonstrate that after introducing Eu^2+^ into the matrix lattice, it exhibits an emission band centered at 508 nm when excited by 365 nm ultraviolet light, which is induced by the 4f^6^5d^1^→4f^7^ transition of Eu^2+^ ions. The optimal doping concentrations of Eu^2+^ and Nd^3+^ were determined to be 2% and 1%, respectively. Based on X-ray diffraction (XRD) analysis, we have found that the physical phase was not altered by the doping of Eu^2+^ and Nd^3+^. Then, we analyzed and compared the quantum yield, fluorescence lifetime, and afterglow decay time of the samples; the co-doped ion Nd^3+^ itself does not emit light, but it can serve as an electron trap center to collect a portion of the electrons produced by the excitation of Eu^2+^, which gradually returns to the ground state after the excitation stops, generating an afterglow luminescence of about 15 s. The quantum yields of SrAl_2_O_4_:Eu^2+^ and SrAl_2_O_4_:Eu^2+^, Nd^3+^ phosphors were 41.59% and 10.10% and the fluorescence lifetimes were 404 ns and 76 ns, respectively. In addition, the E_g_ value of 4.98 eV was determined based on the diffuse reflectance spectra of the material, which closely matches the calculated bandgap value of SrAl_2_O_4_. The material can be combined with polyacrylic acid to create optical anti-counterfeiting ink, and the butterfly and ladybug patterns were effectively printed through screen printing; this demonstrates the potential use of phosphor in the realm of anti-counterfeiting printing.

## 1. Introduction

Doping of nanomaterials with rare earth ions can change their morphology, crystalline phase, size, and electronic configuration, endowing nanomaterials with excellent optical, electrical, and magnetic characteristics. Rare-earth-doped long afterglow luminescent materials quickly absorb energy and promptly generate light when stimulated by an excitation light source. They continue to emit light for a significant duration even after the removal of the excitation light source [1]. Due to its high initial brightness, long fluorescence lifetime, energy saving, and environmental protection, it has been widely used in bio-imaging [2], catalysis [3], environmental remediation [4], anti-counterfeiting [5], and optical information storage [6]. Among many phosphorescent materials, the crystal structure and material composition of SrAl_2_O_4_ have been extensively investigated, owing to its distinctive properties such as excellent fluorescence performance and long-lasting afterglow [7]. During the early 20th century, ZnS:Cu was used as a long afterglow phosphor, but there were problems such as insufficient brightness, short afterglow time, and not being environmentally friendly. The study of SrAl_2_O_4_ began in 1996 when Japanese scientists first developed green long afterglow phosphors co-doped with Eu^2+^ and Dy^3+^ in SrAl_2_O_4_ [8]. In SrAl_2_O_4_:Eu^2+^, Dy^3+^, the Eu^2+^ serves as a luminescent center and Dy^3+^ serves as a trap center to capture electrons and produce a longer afterglow [9]. Since then, in recent decades, researchers have attempted to prepare a series of rare earth luminescent materials based on SrAl_2_O_4_ by different synthesis methods or doped with different rare earth ions [10]. For example, the sol-gel method was employed to prepare SrAl_2_O_4_:Tb^3+^ green long afterglow phosphor [11]; SrAl_2_O_4_:Ce^3+^ phosphors for white light LEDs were prepared by the high-temperature solid-phase method [12]; SrAl_2_O_4_:Eu^2+^, Dy^3+^ luminescent materials have been synthesized using hydrothermal and surfactant–template methods [13]; SrAl_2_O_4_:Eu^3+^ red phosphor was prepared by the combustion method [14]. In numerous studies, Eu^2+^ is commonly used as the luminescent center of SrAl_2_O_4_ phosphor, and Re^3+^ co-doping improves its luminescence and long afterglow performance [15,16,17,18]. SrAl_2_O_4_:Eu^2+^, Nd^3+^ is one of them, and it has received widespread attention in recent years. Ryu et al. synthesized SrAl_2_O_4_:Eu^2+^, Nd^3+^ luminescent materials by the traditional high-temperature solid-phase method [19], and Marchal et al. prepared SrAl_2_O_4_:Eu^2+^, Nd^3+^ long afterglow luminescent materials for ceramic pigments by the sol-gel method [20], but there are problems such as long reaction time. The preparation of SrAl_2_O_4_:Eu^2+^, Nd^3+^ materials by the single-step solution combustion method can significantly reduce the reaction time and cost.

Counterfeit and shoddy goods are emerging, causing serious economic losses to consumers and enterprises. The counterfeited medicines, food, etc., may even threaten consumers’ health and life safety [21]. In this context, anti-counterfeiting technology breaks through the field of banknotes, checks, bonds, stocks, and other securities, and its application in the field of commodity packaging is becoming more and more extensive, for example, hologram anti-counterfeiting, QR code tracking and traceability, optical anti-counterfeiting, watermark design, preparation of anti-counterfeiting ink [22,23,24,25,26], and so on. Among them, the anti-counterfeiting inks prepared by rare earth long afterglow phosphors are applied in the field of printing anti-counterfeiting labels and other fields because of their bright colors, adjustable brightness, and long fluorescence life [27,28,29,30]. The anti-counterfeiting inks are transparent in color under fluorescent light, absorb energy, start to glow after being stimulated, and still can keep the glow for a period of time after withdrawing the stimulation source, which greatly increases the security and feasibility of anti-counterfeit packaging.

In this study, SrAl_2_O_4_:Eu^2+^, Nd^3+^ long afterglow phosphors with optimal doping concentration and reaction temperature were prepared by the combustion method, and we conducted a comprehensive analysis of their photoluminescence, bandgap, crystal structures, elemental compositions, elemental valence, fluorescence lifetimes, and quantum yields. The compound SrAl_2_O_4_:Eu^2+^, Nd^3+^ was subsequently synthesized for the purpose of creating ink that can be used for anti-counterfeiting measures. This ink was then applied to the packaging using screen printing techniques, showcasing the material’s capability for anti-counterfeiting applications.

## 2. Experimental Methods

### 2.1. Materials and Preparation

The combustion method was employed to prepare a series of Sr_1−x−y_Al_2_O_4_:xEu^2+^, yNd^3+^ (x = 0, 0.01, 0.02, 0.03, 0.05, 0.08; y = 0.005, 0.01, 0.02, 0.03, 0.05, 0.08) phosphors. Eu_2_O_3_ (purity: 99.99%), Al_2_O_3_ (purity: 99.99%), Sr_2_CO_3_ (purity: 99.99%), Nd_2_O_3_ (purity: 99.99%), HNO_3_ (purity: 80%), and urea (purity: 99.99%) were utilized as the raw material, all of which were acquired from Tianjin Chemical Reagent Factory. The chemicals can be directly used for experiments without any additional drying or purifying steps.

### 2.2. Synthesis of Materials

Initially, precise quantities of nitric acid (HNO_3_) and purified water were introduced to Eu_2_O_3_, Al_2_O_3_, Sr_2_CO_3_, and Nd_2_O_3_ to prepare Sr(NO_3_)_2_ (0.5 mmol/mL), Al(NO_3_)_3_ (1 mmol/mL), Eu(NO_3_)_3_ (0.1 mmol/mL), and Nd(NO_3_)_3_ (0.5 mmol/mL) solutions, respectively. Subsequently, we used a pipette to measure a specific quantity of reagent into a crucible in accordance with the stoichiometric ratio. Then, 2.2 g of urea particles was added as a reducing and combustion agent and thoroughly mixed until the urea had completely dissolved. Following that, the mixture was placed in a muffle furnace that had been preheated, and it was permitted to burn completely. After approximately 5 min, we extracted the substance and obtained a white solid that was loose and distended. Ultimately, the temperature of the sample was lowered to the ambient temperature of the room, and the substance was ground for a duration of 15–30 min until it became a fine powder, and it was transferred into a test tube for later testing.

The concentrations of Eu^2+^ in the SrAl_2_O_4_:Eu^2+^ material that was synthesized are 0%, 1%, 2%, 3%, 5%, and 8%, respectively. The concentrations of Nd^3+^ in SrAl_2_O_4_:0.02Eu^2+^ and Nd^3+^ materials are 0.5%, 1%, 2%, 3%, 5%, and 8%, respectively. Following the determination of the optimal ion doping concentration, the same experimental method was employed to prepare the samples at reaction temperatures of 500 °C, 600 °C, 700 °C, 800 °C, and 900 °C in order to identify the optimal reaction temperature.

### 2.3. Anti-Counterfeit Ink Preparation

The SrAl_2_O_4_:Eu^2+^, Nd^3+^ phosphor, after preparation, was introduced into a mixture containing ethanol and polyacrylic acid. The amount of phosphor added was adjusted to attain a viscosity appropriate for screen printing. A glass rod was employed to stir the mixture continuously, ensuring the even dispersion of the phosphor in the solution, resulting in the creation of the anti-counterfeit ink. Figure 1 shows a schematic diagram illustrating the process of producing phosphor and anti-counterfeit ink, as well as screen printing.

### 2.4. Characterizing Instruments

The powder X-ray diffractometer (D/max 2200PC by Rigaku, The Woodlands, TX, USA) was used to record the crystal structures of the synthesized samples, which ranged from 10° to 70°, with Cu Kα radiation (λ = 1.54 Å). The binding energy was determined using X-ray photoelectron spectroscopy (XPS, Monochromatic AI Kα(hv = 1486.6 eV) ESCALAB 250 XI, 150W, 650 μm, Waltham, MA, USA). The phosphor morphology was examined using scanning electron microscopy (Quanta 250 FEG, Hitachi, Tokyo, Japan). The elemental compositions and contents were determined using a scanning electron microscope (Thermo Fisher Scientific, Waltham, MA, USA). A fluorescence spectrometer (F4700, Hitachi, Japan) was utilized to conduct photoluminescence spectroscopy (PL) and photoluminescence excitation spectroscopy (PLE). The UV diffuse reflectance absorption spectra were obtained using a UV-3600 Ultraviolet-Visible Near-Infrared Spectrophotometer (Shimadzu Corporation, Kyoto, Japan). The quantum yield, fluorescence lifetime, and afterglow decay curves of phosphors were quantified using a transient steady-state fluorescence spectrometer (Hamamatsu Photonics Quantaurus-Tau C16361-2, Chiyoda, Japan).

## 3. Results and Discussion

### 3.1. XRD Structure Analysis

X-ray diffraction was employed to investigate the physical phase composition and purity of SrAl_2_O_4_ phosphors at varying ion doping concentrations and reaction temperatures. Figure 2a shows the x-ray diffraction (XRD) structure of the SrAl_2_O_4_:0.02Eu^2+^, xNd^3+^ nanophosphors (x = 0.005, 0.01, 0.02, 0.03, 0.05, 0.08) synthesized by calcination at 600 °C and standard cards of the SrAl_2_O_4_ (PDF#97-016-0296). We can observe that the XRD profile exhibits distinct peaks, and that these peaks closely correspond to the data recorded in the standard cards of the SrAl_2_O_4_ (PDF#97-016-0296). The lattice constant of the synthesized sample was measured as a = 8.450Å, b = 8.752Å, c = 5.109Å, β = 93.4° using the JADE 9.0 software. The presence of these peaks can be ascribed to the X-ray reflections originating from the monoclinic SrAl_2_O_4_ crystallographic planes (011), (121), (−211), (221), (210), and (031). Figure 2b shows the XRD patterns of SrAl_2_O_4_:0.02Eu^2+^, 0.01Nd^3+^ nanophosphors obtained by calcination at different temperatures. The diffraction peaks’ positions are essentially unchanged in comparison to Figure 2a, suggesting that the crystal phase of the SrAl_2_O_4_ host is not affected by the co-doping of Eu^2+^ and Nd^3+^ ions.

We also performed Rietveld refinement analysis of XRD data of SrAl_2_O_4_:0.02Eu^2+^, 0.01Nd^3+^ phosphors obtained by calcination at 600 °C using the General Structure Analysis System-II (GSAS-2) software. As shown in Figure 2c, it can be found that the observed data points and the calculated data curves are in good agreement, and no additional impurity peaks are generated. For the SrAl_2_O_4_:0.02Eu^2+^, 0.01Nd^3+^ phosphors, R*_wp_* = 7.04%, and R*_p_* = 4.91% represent the physical and mathematical fit between the sample results and the standard card data, both of which are less than 10%, indicating a favorable curve fitting. From this, we can conclude that the prepared samples are still in single-phase structure with high purity and have no impurities. In addition, Eu^2+^ and Nd^3+^ in the material have a propensity to substitute the Sr^2+^ sites in the main lattice of SrAl_2_O_4_ [31,32,33,34].

In addition, based on the data provided by the XRD pattern, we used Origin 2022 software to fit the XRD pattern using the Gaussian formula, obtained the peak position radians and widths, and calculated the grain size of the sample using the famous Scherrer formula [35]:(1)$$D=\frac{k\lambda }{\beta \cos\theta }$$
where *k* is Scherrer constant with a value of 0.89, *λ* is the wavelength of the radiation with a value of 0.1546, *β* and *θ* represent the FWHM (radians) and the position of the peak, respectively, which have been fitted and calculated by Origin software, and D is the grain size. The average grain sizes of the samples prepared at different temperatures (500–900 °C) were calculated to be 30.593, 33.846, 30.702, 31.227, and 31.338 nm, respectively. In addition, we also calculated the lattice spacing, micro-strain, and dislocation density of the samples, as shown in Table 1.

### 3.2. Elemental Analysis

Since the luminescence efficacy of the materials is directly influenced by the valence states of the doped elements Eu and Nd, we measured the binding energies of the elements in the prepared SrAl_2_O_4_:0.02Eu^2+^, 0.01Nd^3+^ luminescent materials by using X-ray photoelectron spectroscopy (XPS), and conducted comparative analyses to determine the valence states of the doped elements. The XPS measurements, as illustrated in Figure 3a, demonstrate binding energies that correspond to Sr 3d, Al 2p, O 1s, C 1s, Eu 3d, and Nd 3d. Figure 3b displays the high-resolution (HR) XPS spectra of Sr 3d. The peak binding energies of Sr 3d are observed to be 133.60 eV and 135.24 eV, which is consistent with that of Sr 3d_5/2_ and Sr 3d_3/2_ [36,37], respectively. This is related to the occupancy of Sr elements in SrAl_2_O_4_ [38]. Figure 3e illustrates the binding energy of Eu 3d peaks at 1125.95 eV for Eu^2+^ 3d_5/2_ and at 1155.06 eV for Eu^2+^ 3d_3/2_, which demonstrates that the Eu ions in the prepared SrAl_2_O_4_:0.02Eu^2+^, 0.01Nd^3+^ luminescent materials are in the divalent oxidation state [39,40]. Figure 3f shows that the peak binding energies of Nd 3d are 977.83 eV and 1000.31 eV, which is consistent with that of Nd^3+^ 3d_5/2_ and Nd^3+^ 3d_3/2_ [41,42,43]. In addition, we employed Avantage 5.9 software to analyze and calculate the XPS data of the substance, resulting in the determination of the atomic ratios of SrAl_2_O_4_:Eu^2+^, Nd^3+^ powders. The measured data are incongruous with the EDS test results, attributable to the disparate testing principles employed by the two methods, as shown in Table 2. This also proves the difference in element distribution between the surface and volume of the sample, which is caused by the combustion method. This reaction may result in an uneven distribution of elements within the material.

### 3.3. SEM Analysis

Scanning electron microscopy (SEM) was employed to examine the impact of temperature on the surface morphology of the samples. Figure 4a–e depict the SEM images of SrAl_2_O_4_:0.02Eu^2+^, 0.01Nd^3+^ luminescent materials produced at different reaction temperatures; the magnification is 10,000×, with an average scale size of approximately 1 µm. From the figure, it can be observed that there are many cracks and pores on the surface of the sample, which are caused by the violent reaction that occurs in a short period of time during the combustion method [17,44]. During the combustion process, many gases are released, leading to this phenomenon. Similarly, we observed the aggregation of sample particles into clustered structures, which reflects the intrinsic behavior of combustion. The particles in the powder are uneven and their sizes are not exactly the same; this could be attributed to the inhomogeneous temperature distribution of the sample during the reaction [45], uneven mass flow distribution, and differences in the degree of grinding [46]. We can also observe that the sample prepared at 500 °C has larger pore gaps. When the reaction temperature reaches 600 °C, the pores between the sample particles gradually become smaller, and the particle size is relatively regular and uniform. As the temperature further increases, the pores between the sample particles become larger and the size becomes more uneven. The structural variations can affect the absorption and emission process of light, which explains why the sample prepared at 600 °C has the highest luminescence intensity. Moreover, different reaction temperatures can also affect the distribution of dopants, thereby affecting luminescence [47,48,49].

Furthermore, the presence of the elements Sr, Al, O, Eu, and Nd is confirmed by the Energy Dispersive X-ray Spectroscopy (EDS) spectra of SrAl_2_O_4_:0.02Eu^2+^, 0.01Nd^3+^, as depicted in Figure 4f. The elemental composition of the material and the energy spectrum intensity signals of each element were determined by measuring small areas of the sample. The chemical composition of SrAl_2_O_4_:0.02Eu^2+^, 0.01Nd^3+^ is essentially consistent with the spectral intensity signals of Sr, Al, O, Eu, and Nd. The proportion of Eu and Nd elements in the sample is slightly higher than the standard value of the sample chemical formula, possibly as a result of the combustion method used in this study. This method completes the doping process in a shorter amount of time through solution combustion, which results in a more violent reaction. This reaction may result in an uneven distribution of elements within the material. The EDS detected the presence of Eu and Nd, indicating that Eu and Nd were successfully doped into the host material.

### 3.4. Photoluminescence Analysis

Figure 5 displays the spectra of photoluminescence (PL) and photoluminescence excitation (PLE) for SrAl_2_O_4_ luminescent materials. These materials were produced using various ion doping concentrations and reaction temperatures. Figure 5 depicts the fluorescence emission spectra of Sr_1-x_Al_2_O_4_:xEu^2+^ (x = 0, 0.01, 0.02, 0.03, 0.05, 0.08) at 365 nm excitation wavelength, and the combustion temperature is 600 °C. It is evident that the luminescence intensity of the samples increased progressively as the concentration of Eu^2+^ increased, and peaked at 3% Eu^2+^ concentration, after which the luminescence intensity gradually decreased. The spectrum exhibits a wide emission band in the green region, with a peak at 508 nm. An increase in the concentration of Eu^2+^ does not change the peak location of the emission spectrum of the luminescent material, but it does impact the luminescence intensity of the sample, which is attributable to the transition of the Eu^2+^ ion from the high-energy state 4f^6^5d to the low-energy state 4f^7^. Similarly, Figure 5b displays the fluorescence emission spectra of Sr_0.98-y_Al_2_O_4_:0.02Eu^2+^, yNd^3+^ (y = 0.005, 0.01, 0.02, 0.03, 0.05, 0.08) at 365 nm excitation wavelength with a reaction temperature of 600 °C. When the doping concentration of Nd^3+^ is 1%, the luminescence intensity of the sample is at its highest, and the maximal emission peak is also at 508 nm, as illustrated in the figure. After determining the optimal ion doping concentration, we prepared SrAl_2_O_4_:0.02Eu^2+^, 0.01Nd^3+^ luminescent materials with different temperatures, and their photoluminescence spectra are shown in Figure 5c. The luminescence intensity of the material is significantly influenced by the reaction temperature; the luminescence intensity of the sample is at its highest when the reaction temperature is 600 °C, as illustrated in the figure. The photoluminescence excitation (PLE) spectra of the samples after excitation are depicted in Figure 5d. The PLE spectra exhibit a paragraph absorption band that is centered at 365 nm, consistent with the emission spectra, and corresponds to the 4f^7^→4f^6^5d transition of the Eu^2+^.

The monoclinic crystal structure of SrAl_2_O_4_ contains two distinct Sr sites, Sr1 and Sr2, which are essentially identical in size [50]. During the doping process, given the comparable ionic radii of Eu^2+^ and Sr^2+^, Eu^2+^ tends to substitute for Sr^2+^ in SrAl_2_O_4_. Eu^2+^ at both sites undergoes a 4f→5d transition, resulting in luminescence [51]. In addition, the samples doped with Eu^2+^ and Nd^3+^ did not exhibit any emission properties from the Nd^3+^ ions. This indicates that Nd^3+^ may not function as a source of luminescence, but instead may serve as a center for trapping. Nd^3+^ releases stored energy for sustained luminescence after cessation of excitation [52]. An increase in the concentration of Nd^3+^ ions leads to the generation of a greater number of electron traps, resulting in an increased capacity to store electrons. The release of these electrons occurs gradually following the cessation of excitation, resulting in an increase in the intensity of the afterglow. However, beyond a certain concentration, the high density of electron traps will result in a concentration quenching, and the brightness of the afterglow will be reduced [53,54,55,56].

The chromaticity diagram can be used to represent the chromatic properties of all colors. According to the spectral trajectory, the color coordinate values corresponding to the spectral tri-stimulus values of the CIE1964 supplemental standard chromaticity observer can be plotted on a CIE1964xy color coordinate chart, as shown in the horseshoe curve in Figure 5e. The chromaticity diagram can be used to represent the chromaticity characteristics of all colors. The dots on the spectral track represent the spectral colors of different wavelengths, which are the most saturated colors, and the closer to the center of the chromaticity diagram (the white dot), the less saturated the colors. Different angles of rotation around the center of the chromaticity diagram correspond to different wavelengths of the spectral colors, representing different shades. The CIE chromaticity coordinates *x* and *y* can be determined using the following equation [57]:(2)$$x=\frac{X}{X+Y+Z}$$
(3)$$y=\frac{Y}{X+Y+Z}$$
where *X*, *Y*, and *Z* are the CIE tristimulus values. The CIE1964xy color coordinate values were obtained by calculating the spectral data of the samples with different Nd^3+^ ion doping concentrations, as shown in Figure 5e. It can be seen that the color is green at an Nd^3+^ ion concentration of 0.01 and the coordinates are (x = 0.189, y = 0.437); as the concentration increases, the color shifts to blue, and when the doping concentration is 0.08, the coordinates are (x = 0.168, y = 0.144).

Figure 5f illustrates the luminescent mechanism of Eu^2+^ when stimulated by 365 nm excitation. Upon exposure to UV lamp irradiation, the material absorbs photon energy, leading to the excitation of electrons and subsequent 4f-5d transitions. Upon cessation of excitation, the ion undergoes a transition from the 5d excited state to the ground state of the 4f orbital, resulting in the emission of energy and the generation of light. The emission wavelength is determined by the energy level configuration of the ion and its chemical surroundings in the material. When rare earth ions transition from the excited state to the ground state, they release photons. The emitted light often has a longer wavelength than the absorbed light, a phenomenon known as the Stokes shift.

### 3.5. UV–Vis Diffuse Reflectance

The UV diffuse reflectance spectra of SrAl_2_O_4_:0.02Eu^2+^ and SrAl_2_O_4_:0.02Eu^2+^, 0.01Nd^3+^ phosphors, which were prepared at 600 °C, are depicted in Figure 6a and Figure 6b, respectively, within the 200–800 nm range. The absorption characteristics and bandgap values of SrAl_2_O_4_:Eu^2+^, Nd^3+^ were determined through analyzing and calculating its absorption spectrum. The wide absorption band generated by the material at 300–400 nm corresponds to the 4f→5d transition of Eu^2+^, as shown in the figure. The luminescence performance of the material is strongly correlated with its bandgap value, and it has been established through a review of the existing literature that SrAl_2_O_4_ possesses a direct bandgap. A spectrum was generated using the Tauc formula to determine the bandgap value of the material, based on the measurement data of the material’s UV diffuse reflectance spectrum. The Tauc formula can be expressed as [58,59]:(4)$${\left(\alpha h\nu \right)}^{2}=A(hv-E_{g})$$
where *A* is constant, *hv* is the photon energy, *α* is the optical absorption coefficient. The insets in Figure 6a,b show the Tauc curves for the two materials, reflecting the (*αhν*)^2^ versus the energy. The bandgap of synthesized SrAl_2_O_4_:0.02Eu^2+^ and SrAl_2_O_4_:0.02Eu^2+^, 0.01Nd^3+^ was calculated to be 5.23 eV and 4.98 eV, respectively.

### 3.6. Photoluminescence Quantum Yield (PLQY)

Photoluminescence quantum yield (*PLQY*) is the ratio of the number of photons emitting secondary radiated fluorescence to the number of photons absorbing primary radiated photons of excitation light per unit time. This concept is used to quantify the ability of a substance to emit fluorescence after absorbing light energy and is an essential parameter for luminescent materials. The following equation can be used to calculate the quantum yield by combining the measured number of emitted photons (*ε*) and the number of absorbed photons (*α*) of the sample [60]:(5)$$PLQY=\varepsilon /\alpha =\frac{S_{em}}{S_{0}-S}$$
where *S_em_* represents the total intensity of the emitted light from the phosphor, and S_0_ and S represent the total intensity of the scattered light from the reference background plate and the phosphor, respectively. Figure 7 shows a schematic diagram of the quantum yield of three phosphors, SrAl_2_O_4_, SrAl_2_O_4_:0.02Eu^2+^, and SrAl_2_O_4_:0.02Eu^2+^, 0.01Nd^3+^, under the excitation light of 365 nm, with the red color representing the samples and the black color representing the reference white board, and the difference in the area between the scatter range (355 nm–375 nm) and the emission range (400 nm–800 nm) is the quantum yield.

The plots show quantum yields of 5.05%, 41.59%, and 10.10% for SrAl_2_O_4_, SrAl_2_O_4_:0.02Eu^2+^, and SrAl_2_O_4_:0.02Eu^2+^, 0.01Nd^3+^, respectively. It is evident that the doping of rare earth ions greatly enhances the quantum yield of the material and improves its luminescence ability, indicating that the special structure of Eu^2+^ ions can effectively realize the transfer of energy, which can substantially improve the quantum yield [61,62]. The quantum yield of the SrAl_2_O_4_:0.02Eu^2+^ sample is slightly lower [63], and we will focus on this aspect in our future work. The low quantum yields of the sample doped with Nd^3+^ ions can be attributed to the introduction of non-radiative dissipation channels by the addition of Nd^3+^ ions. These ions have the ability to absorb a portion of the energy into their energy levels, resulting in a loss in quantum yield [64,65].

### 3.7. Fluorescence Lifetime and Afterglow Decay

The fluorescence lifetime is the duration of time required for the fluorescence intensity of a material to decrease to 1/e of its initial value after the excitation light source is removed. Figure 8 shows the fluorescence decay curves and the fitted curves of SrAl_2_O_4_:0.02Eu^2+^, yNd^3+^ (y = 0.005, 0.01, 0.02, 0.03, 0.05) at 365 nm excitation wavelength, including the initial rapid decay and the subsequent slow decay. A third-order exponential function can be used to estimate the fluorescence decay curve [66]:(6)$$I\left(t\right)=I_{0}+A_{1}e^{-\frac{t}{\tau 1}}+I_{0}+A_{2}e^{-\frac{t}{\tau 2}}+A_{3}e^{-\frac{t}{\tau 3}}$$
where *I* is the phosphorescence intensity; *I*_0_, *A*_1_, *A*_2_, and *A*_3_ are constants; t is the time; *τ*_1_, *τ*_2_, and *τ*_3_ are the exponential component decay times. The above values can be calculated by Origin software, as shown in Table 3. The average decay time τ* of SrAl_2_O_4_ under different Nd^3+^ doping concentrations can be obtained using the following equation [67]:(7)$$\tau_{ave}=\frac{A_{1}{\tau_{1}}^{2}+A_{2}{\tau_{2}}^{2}+A_{3}{\tau_{3}}^{2}}{A_{1}\tau_{1}+A_{2}\tau_{2}+A_{3}\tau_{3}}$$

The average decay times *τ** of Sr_0_._98-y_Al_2_O_4_:0.02Eu^2+^, yNd^3+^ (y = 0, 0.005, 0.01, 0.02, 0.03, 0.05) were calculated from the above equation as 404 ns, 221 ns, 76 ns, 46 ns, 119 ns, and 135 ns, respectively. The fluorescence lifetime of the SrAl_2_O_4_:0.02Eu^2+^ sample is similar to that of the sample prepared by the high-temperature solid-phase method [68]. From Figure 8, the longest fluorescence lifetime is observed in the SrAl_2_O_4_:0.02Eu^2+^ phosphor, which begins to decrease as the concentration of Nd^3+^ doping increases; this is due to the formation of trap centers when Nd^3+^ replaces Sr^2+^ in the matrix, capturing some electrons. The phosphor’s fluorescence lifetime begins to decay as the concentration of Nd^3+^ ions continues to rise, which may be related to the high density of electron traps, which produces the concentration quenching phenomenon.

To conduct a more detailed examination of the afterglow properties of the phosphors, we analyzed the decay curves of the afterglow for SrAl_2_O_4_:0.02Eu^2+^, 0.01Nd^3+^, and SrAl_2_O_4_:0.02Eu^2+^, 0.02Nd^3+^ at an excitation wavelength of 365 nm. The fluorescence decay curves and the calculation of the average afterglow duration are based on Equations (5) and (6) for the third-order exponential fitting, and the parameters are shown in Table 4. Similarly, the decay processes of the samples are categorized into two types: early rapid decay and subsequent gradual decay. The rapid decay process dominates the intensity and the slow decay process is also known as the long afterglow luminescence process [69]. As depicted in Figure 9, the average afterglow duration τ* of SrAl_2_O_4_:0.02Eu^2+^, 0.01Nd^3+^ is 13 s, while the average afterglow duration of SrAl_2_O_4_:0.02Eu^2+^, 0.02Nd^3+^ is 8 s. The reason why the afterglow is produced for a shorter period of time is because the energy is dissipated too quickly during the energy transfer process. This phosphor can rapidly transfer energy to the surrounding area through non-radiant energy transfer, which greatly improves the efficiency. Based on this property, these phosphors are well suited for fluorescent signal detection as well as anti-counterfeiting applications [70,71].

### 3.8. Anti-Counterfeiting Application

In order to confirm the potential application of the prepared phosphors in anti-counterfeiting printing, we combined the SrAl_2_O_4_:0.02Eu^2+^ and SrAl_2_O_4_:0.02Eu^2+^, 0.01Nd^3+^ fluorescent powders separately with polyacrylic acid and ethanol to create two types of anti-counterfeiting ink. The butterfly and ladybug patterns were printed through screen printing. As depicted in Figure 10, it can be observed that the pattern does not show color under fluorescent light, and the printed pattern can be clearly seen under the irradiation of a 365 nm UV lamp. The color of SrAl_2_O_4_:0.02Eu^2+^ fluorescent ink disappears after turning off the UV lamp, but the SrAl_2_O_4_:0.02Eu^2+^, 0.01Nd^3+^ fluorescent ink can continue to emit light for a period of time. Therefore, we can consider the prepared luminescent material as having a significant potential for application in the domain of anti-counterfeit printing.

## 4. Conclusions

In conclusion, the combustion method was employed to produce a series of SrAl_2_O_4_:0.02Eu^2+^ and SrAl_2_O_4_:0.02Eu^2+^, 0.01Nd^3+^ phosphors. The successful doping of Eu^2+^ and Nd^3+^ ions into the SrAl_2_O_4_ host was demonstrated by XRD, EDS, and XPS characterization tests. Photoluminescence tests were carried out on the phosphors, and the results showed that both SrAl_2_O_4_:Eu^2+^ and SrAl_2_O_4_:Eu^2+^, Nd^3+^ exhibited green luminescence when exposed to 365 nm excitation light, with a peak emission at 508 nm, which is caused by the 4f^6^5d^1^→4f^7^ transition of Eu^2+^. The fluorescence lifetimes of SrAl_2_O_4_:0.02Eu^2+^ and SrAl_2_O_4_:0.02Eu^2+^, 0.01Nd^3+^ were 404 ns and 76 ns and the quantum yields were 41.59% and 10.10%. The Nd^3+^-doped samples showed strong afterglow performance with an afterglow lifetime of 13s. The bandgap values of the samples were measured by plotting Tauc curves from UV diffuse reflectance spectra, which matched the calculated bandgap values. SrAl_2_O_4_:0.02Eu^2+^ and SrAl_2_O_4_:0.02Eu^2+^, 0.01Nd^3+^ phosphors were prepared for anti-counterfeiting inks, and better anti-counterfeiting effects were obtained by printing the patterns by screen printing.

## Figures and Tables

**Figure 1 nanomaterials-14-01265-f001:**
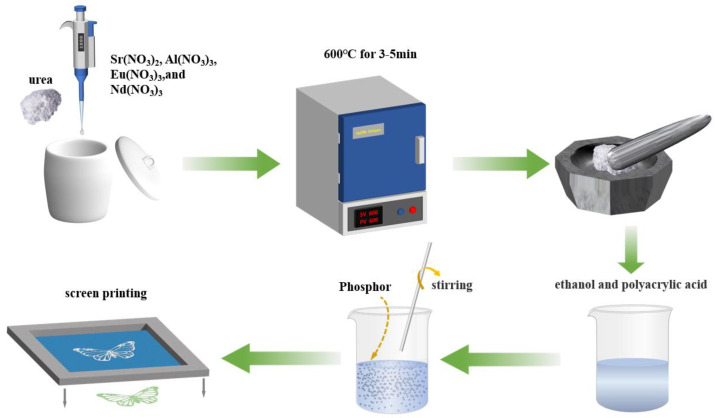
Flow process diagram of phosphor preparation and screen printing.

**Figure 2 nanomaterials-14-01265-f002:**
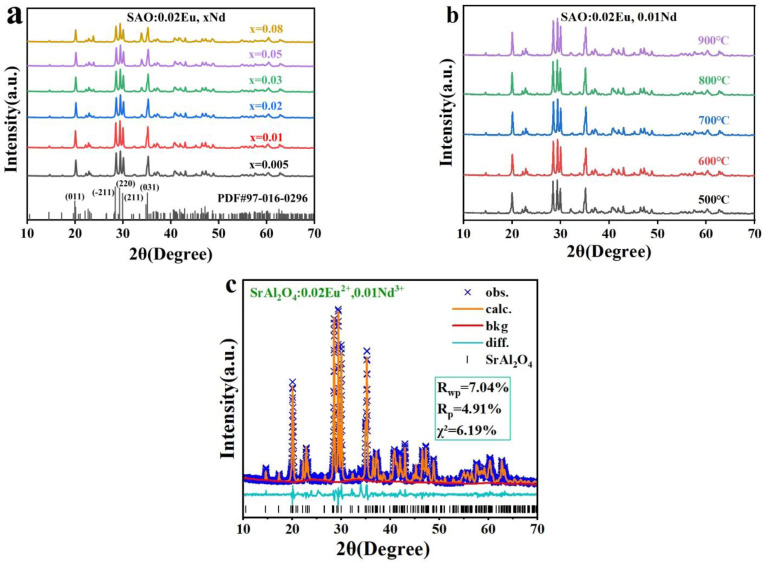
(**a**) The XRD structures of SrAl_2_O_4_:0.02Eu^2+^, xNd^3+^ (x = 0.005, 0.01, 0.02, 0.03, 0.05, 0.08) phosphors. (**b**) The XRD structures of SrAl_2_O_4_:0.02Eu^2+^, 0.01Nd^3+^ at different sintering temperatures (500, 600, 700, 800, 900 °C). (**c**) Rietveld refinement of the SrAl_2_O_4_:0.02Eu^2+^, 0.01Nd^3+^ phosphor.

**Figure 3 nanomaterials-14-01265-f003:**
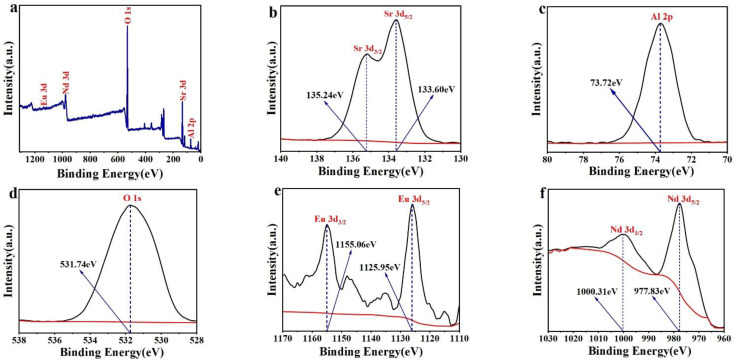
(**a**) The XPS spectra of SrAl_2_O_4_:Eu^2+^, Nd^3+^; (**b**–**f**) the high-resolution (HR) XPS spectra of Sr 3d, Al 2p, O 1s, Eu 3d, and Nd 3d.

**Figure 4 nanomaterials-14-01265-f004:**
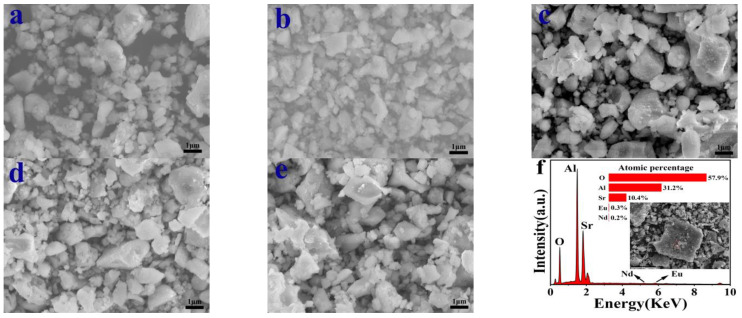
SEM images of SrAl_2_O_4_:0.02Eu^2+^, 0.01Nd^3+^ synthesized at (**a**) 500 °C, (**b**) 600 °C, (**c**) 700 °C, (**d**) 800 °C, (**e**) 900 °C. (**f**) The EDS of SrAl_2_O_4_:0.02Eu^2+^, 0.01Nd^3+^ powder prepared at 600 °C.

**Figure 5 nanomaterials-14-01265-f005:**
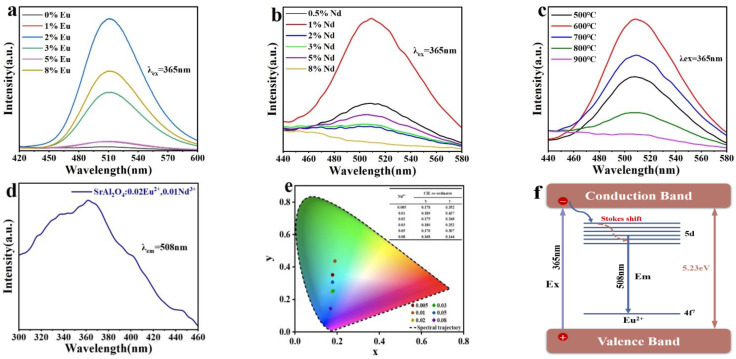
(**a**,**b**) The emission spectra (λ_ex_ = 365 nm) of the SrAl_2_O_4_:xEu^2+^ (x = 0, 0.01, 0.02, 0.03, 0.05, 0.08) and SrAl_2_O_4_:0.02Eu^2+^, yNd^3+^ (y = 0.005, 0.01, 0.02, 0.03, 0.05, 0.08); (**c**) the emission spectra (λ_ex_ = 365 nm) of the SrAl_2_O_4_:0.02Eu^2+^, 0.01Nd^3+^ at different sintering temperatures (500–900 °C); (**d**) photoluminescence excitation (PLE) spectra of SrAl_2_O_4_:0.02Eu^2+^, 0.01Nd^3+^ detected at 508 nm; (**e**) CIE chromaticity plot of the SrAl_2_O_4_:0.02Eu^2+^, 0.01Nd^3+^ stimulated by 365 nm UV light; (**f**) the luminescent mechanism of Eu^2+^ when stimulated by 365 nm excitation.

**Figure 6 nanomaterials-14-01265-f006:**
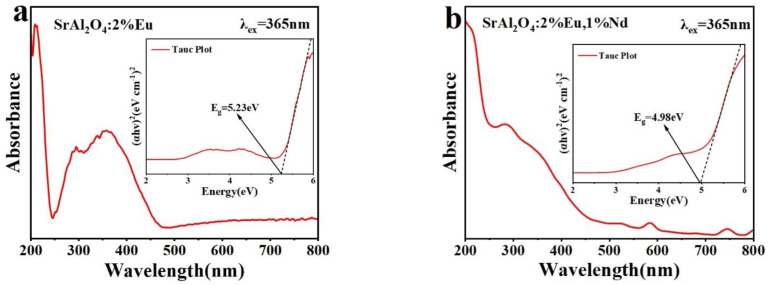
UV absorption spectrum and the bandgap of (**a**) SrAl_2_O_4_:0.02Eu^2+^; (**b**) SrAl_2_O_4_:0.02Eu^2+^, 0.01Nd^3+^.

**Figure 7 nanomaterials-14-01265-f007:**
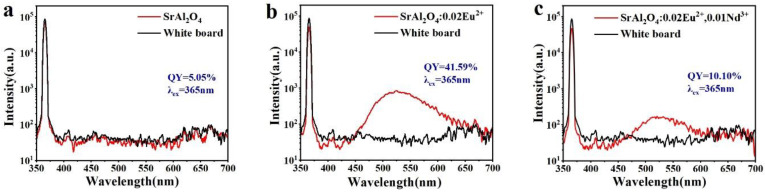
The quantum yield curves of (**a**) SrAl_2_O_4_; (**b**) SrAl_2_O_4_:0.02Eu^2+^; and (**c**) SrAl_2_O_4_:0.02Eu^2+^, 0.01Nd^3+^.

**Figure 8 nanomaterials-14-01265-f008:**
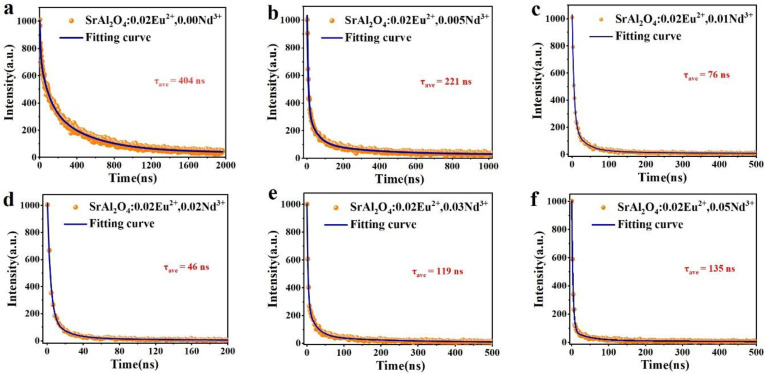
(**a**–**f**) The fluorescence lifetime decay curves and fitting curves of SrAl_2_O_4_:0.02Eu^2+^, xNd^3+^ (x = 0, 0.005, 0.01, 0.02, 0.03, 0.05) phosphors.

**Figure 9 nanomaterials-14-01265-f009:**
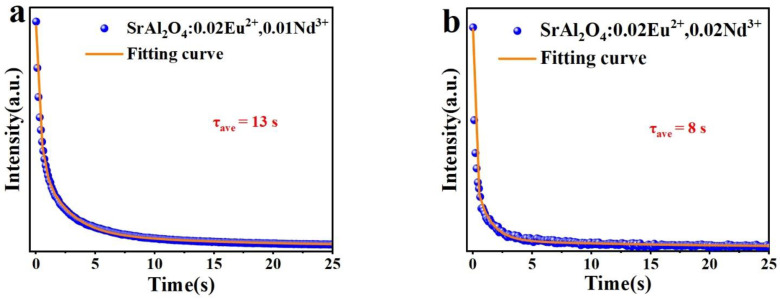
The afterglow decay curves and fitting curves of (**a**) SrAl_2_O_4_:0.02Eu^2+^, 0.01Nd^3+^ and (**b**) SrAl_2_O_4_:0.02Eu^2+^, 0.02Nd^3+^ phosphors after exposure to a 365 nm UV light for 3 min.

**Figure 10 nanomaterials-14-01265-f010:**
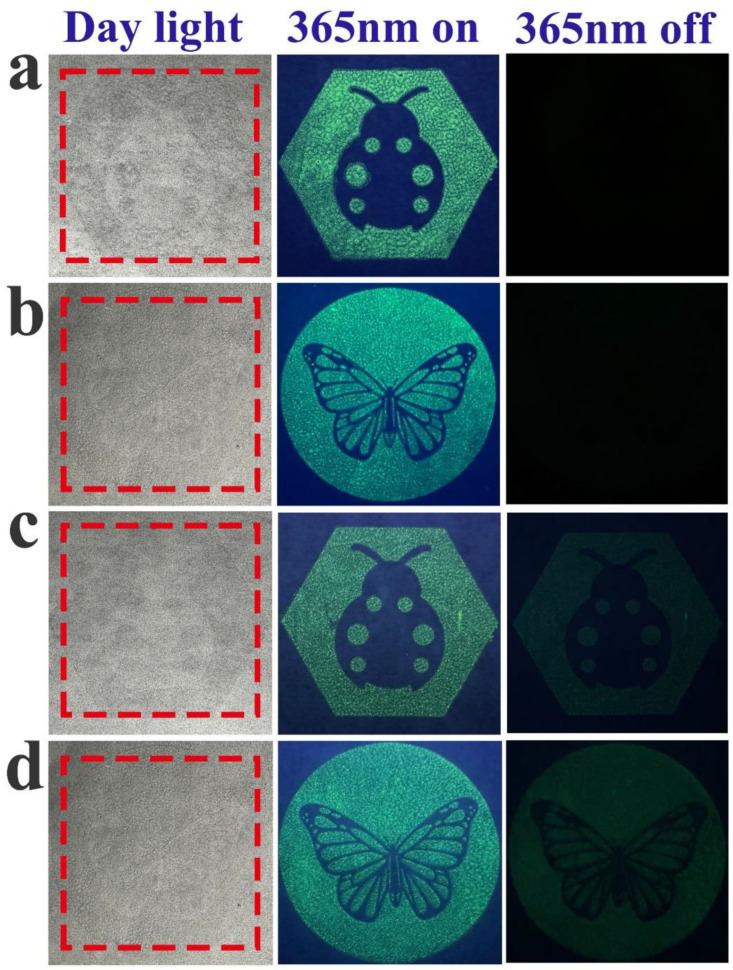
The (**a**) ladybug and (**b**) butterfly patterns printed with SrAl_2_O_4_:0.02Eu^2+^ ink; the (**c**) ladybug and (**d**) butterfly patterns printed with SrAl_2_O_4_:0.02Eu^2+^, 0.01Nd^3+^ ink.

**Table 1 nanomaterials-14-01265-t001:** XRD data of SrAl_2_O_4_:Eu^2+^, Nd^3+^ phosphor.

2θ	FWHM (β)	Lattic Spacing (d)	Intensity (I)	hkl	Crystallite Size (D)	Dislocation Density (δ)	Micro-Strain (ε)
20.066	0.257	0.442	60	011	31.010	1.040	6.345
28.487	0.222	0.313	95	−211	36.589	0.747	3.808
29.370	0.206	0.304	100	220	39.424	0.643	3.430
30.002	0.204	0.298	79	211	39.775	0.632	3.330
35.147	0.367	0.255	67	031	22.430	1.988	5.062

**Table 2 nanomaterials-14-01265-t002:** The atomic ratio of SrAl_2_O_4_:Eu^2+^, Nd^3+^ phosphor.

Method	Atomic%				
	Sr	Al	O	Eu	Nd
XPS	9.90	20.16	66.50	0.77	2.67
EDS	10.40	31.20	57.90	0.30	0.20

**Table 3 nanomaterials-14-01265-t003:** The average fluorescence lifetime of SrAl_2_O_4_:0.02Eu^2+^, xNd^3+^ (x = 0, 0.005, 0.01, 0.02, 0.03, 0.05) phosphors.

Nd^3+^	Decay Lifetimes (ns)						
	A_1_	τ_1_	A_2_	τ_2_	A_3_	τ_3_	τ*
0.00	256	8	343	96	367	468	404
0.005	668	6	250	48	88	325	221
0.01	804	5	195	30	24	193	76
0.02	169	17	825	3	22	121	46
0.03	719	3	231	22	48	196	119
0.05	69	41	919	3	13	301	135

**Table 4 nanomaterials-14-01265-t004:** Afterglow decay time for SrAl_2_O_4_:0.02Eu^2+^, 0.01Nd^3+^ and SrAl_2_O_4_:0.02Eu^2+^, 0.02Nd^3+^ samples.

Nd^3+^	Afterglow Decay Time (s)						
	A_1_	τ_1_	A_2_	τ_2_	A_3_	τ_3_	τ*
0.01	33,392	2	53,015	0	6063	15	8
0.02	709	1	1503	0	85	21	13

## Data Availability

Data is contained within the article.

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
