# Peer review of "Study on the Luminescence Performance and Anti-Counterfeiting Application of Eu2+, Nd3+ Co-Doped SrAl2O4 Phosphor"

_nanomaterials, 2024, doi:10.3390/nano14151265_

Round 1

Reviewer 1 Report

Comments and Suggestions for Authors

1.      1.  The phrase on lines 38-41 can be deleted as not directly related to the content of the article:

«Rare earth elements are a collection of 17 elements, including scandium, yttrium, and 15  lanthanides ranging from lanthanum(La) to lutetium(Lu), that have similar physical and chemical  characteristics. Since the discovery of the first rare earth mineral in 1751, the separation of 17 rare 40 earth elements was not completed until 1909 due to their mostly similar chemical properties[1]» 

2.      In the “XRD structure analysis” section, the listing of 2θ angle values ​​should be removed, and the indices (hkl) should be given in the figure. Instead of unit cell parameters from the literature, one should provide one's own experimental data.

3. What elemental analysis method was used – energy dispersive spectroscopy (EDS) or wavelength-dispersive X-ray spectroscop? What is the error in determining of neodymium and europium? The sensitivity of the method EDS is not sufficient to draw conclusions about the effect of oxygen on the ratio Nd/Eu

Reviewer 2 Report

Comments and Suggestions for Authors

In this investigation Z. Wang and his co-workers investigated structural, morphological and optical properties of SrAl2O4 phosphor co-doped with Eu2+ and Nd3+ with potential application in anti-counterfeiting printing. They also analyzed optimal concentration of doping ions, quantum yield, fluorescence lifetime and afterglow decay time. The manuscript contains a lot of interesting results and I recommend this article to be published in your Journal, after a major revision.

However, some corrections must be taken before the acceptance.

In the introduction part, the authors must give the literature overview of the synthesized material and emphasize the novelty of their work.

Regarding the XRD measurements, since you already did the structural refinement, you should give the table with the basic parameters in order to see their change with the raising of temperature, lattice parameters, crystallite size, strain, etc. From the table, the authors should give the appropriate explanation about the refinement. Since the existence of 2 position in Sr does the dopants have the preference of accommodation?

In XPS measurements it would be interesting to give the quantitative analysis in order to see if there is some mismatch between XPS and EDS analysis, i.e. surface and bulk.

SEM analysis is written very poorly. The authors must do additional figures in much higher magnification and explain the morphology in detail. Also, comments regarding atomic percentage with EDS and XPS are welcome. It is well known that morphology greatly influences the luminescence so this part must be improved.

If the authors can do the TEM measurements it would be interesting to see some mapping, or they can do that with SEM. It would be interesting to see does the temperature influence the distribution of dopants in the samples.

Photoluminescence part is written very good. But, as for lifetime and quantum yield, the authors have to give some reference to compare this sample with the literature.

It should be noted that energy diagram should be implemented.

Reviewer 3 Report

Comments and Suggestions for Authors

The Authors prepared a co-doped Eu, Nd SrAl2O4 phosphor by combustion method, although there are other works with Eu and Nd doping agents, the authors highlight the possibility of producing an anti-counterfeiting ink. The authors made a great effort in carrying out numerous measurements and characterizing the samples. In particular, luminescence was studied. 

Some questions:

-By XRD measurements the Authors claim that the dopants substitute Sr, but there is no variation in the diffraction patterns as a function of dopants concentration and preparation temperature. In my opinion substitutions induce some peak variation, and the effect must be detected in XRD pattern. Moreover, the sintering temperature affect the grain dimensions, it can be calculated by the Scherrer equation. 

-The effect of sintering temperature influences also the emission spectra. Some more explanation will be interesting. In my opinion, looking at figure 4, this effect can depend on the grain dimensions, distribution and/or structural variations. So, it would be interesting at least to perform the mean dimensions measurement.

Moreover, despite the Journal is “Nanomaterials” the dimensions are far to be nanometric.

-Lines 278-281: As a matter of fact, there is a variation in the band gap (fig. 6), otherwise it is not possible to explain the luminescence variation

Round 2

Reviewer 2 Report

Comments and Suggestions for Authors

The revised version of the manuscript can be published in Nanomaterials.